# Positive outcomes among nursing home caregivers in Spain during the COVID-19 pandemic: A qualitative interview study

Macarena Sánchez-Izquierdo[1☯], Rodrigo García-Sánchez[1‡], María Prieto-Ursúa[2☯], Jesús Mateos-Nozal[3‡], Ángela Ordóñez-Carabaño[1☯]*

1 UNINPSI Clinical Psychology Center, and Department of Psychology, Comillas Pontifical University, Madrid, Spain, 2 Department of Psychology, Comillas Pontifical University, Madrid, Spain, 3 Unidad de Geriatría de Atención a Residencias, Servicio de Geriatría, Hospital Universitario Ramón y Cajal (IRYCIS), Madrid, Spain

☯ These authors contributed equally to this work.
‡ RG-S and JM-N also contributed equally to this work.
* aordonez@comillas.edu

## Abstract

The COVID-19 pandemic placed unprecedented strain on professional caregivers in nursing homes. This study aimed to explore the positive outcomes these caregivers experienced amidst the crisis. We conducted 15 semi-structured interviews with nurses, medical staff, and support personnel from two nursing homes in Spain between December 2021 and July 2022. Through qualitative content analysis, we uncovered signs of post-traumatic growth, such as enhanced problem-solving and organizational skills. A strong social support network was key to fostering this growth, easing distress, and strengthening team cohesion. Caregivers expressed a deep sense of gratitude and satisfaction in their roles, finding new meaning in their work, which contributed to resilience and a positive outlook despite ongoing challenges. These findings underscore the importance of robust support systems in caregiving environments, which can not only mitigate stress but also promote personal and professional growth. Further research should focus on sustaining these positive adaptations and developing strategies that enhance resilience and well-being for caregivers, ultimately improving care outcomes for older adults in nursing homes.

## Introduction

The COVID-19 pandemic presented significant global challenges, with older adults being particularly vulnerable to severe outcomes. Among these, residents of nursing homes were one of the most affected groups in terms of infections and deaths, having more severe cases and deaths from SARSCoV-2 infection during the first wave than in any other setting in Spain. During the first wave the number of deaths with confirmed or suspected COVID-19 in nursing homes was almost 20.000 patients [1].

**Data availability statement:** Data is available for replication by contacting the ethical committee [dpo@comillas.edu]. The study was not preregistered.

**Funding:** This work was supported by Comillas Pontifical University through the 2022 call for internal research projects. The research project is entitled "Positive Effects of the COVID Pandemic on Professional Caregivers of Elderly Individuals in Nursing Homes" (PP2022_10).

**Competing interests:** The authors have declared that no competing interests exist.

Moreover, the effects of the crisis were far-reaching, impacting not only the residents but also their families and healthcare staff [2].

In Spain in 2020 there were over 5.000 nursing homes for the care of older adults with more than 380.000 beds, and important differences regarding management (public or private) and size between different regions and centers [3]. In march of that year it was promoted the coordination between nursing homes and the health system for COVID-19 diagnosis and management, and it was enforced prevention measures in these centers, including internal confinements, visitor limitations, and enhanced access controls to protect the highly vulnerable resident population [4].

While necessary for outbreak control, these restrictions introduced operational complexities and required adaptation from all nursing home personnel [5]. Additionally, these restrictive policies sometimes had unintended negative impacts on both residents and their formal and informal caregivers, highlighting the delicate balance required in crisis management within long-term care facilities [6,7].

Moreover, this multi-layered, collaborative response was crucial for effective crisis management [8], as well as different factors from the centers and the residents that constrained the capacity of these facilities to meet the demands of the COVID-19 crisis [6,9]. The collaboration among professionals helped mitigate some adverse effects of the pandemic. Research highlights that collaborative coordination enabled a faster, adaptive response to the health crisis, underscoring the importance of cross-disciplinary cooperation within nursing homes [10]. Positive outcomes of this coordination were observed in improved resident care and reduced hospitalization rates [11].

Despite the significant risks of burnout and mental health deterioration associated with caregiving [12], there is evidence that caregiving can also offer deeply fulfilling experiences. Research indicates that undergoing stress and trauma can lead to a process of rethinking and personal development, suggesting that such experiences can contribute to significant positive growth and changes in perspective, such as posttraumatic growth [13], resilience [14], and professional satisfaction [15]. A study involving Spanish health personnel revealed feelings of pride, satisfaction from assisting others, and professional fulfillment [16]. Similarly, Blanco-Donoso, et al. [17] found that professionals in nursing homes were able to achieve professional satisfaction amidst the COVID-19 crisis.

Caregiving, despite its challenges, can offer positive changes to one's life, with both positive and negative aspects of caregiving coexisting, even during the COVID-19 pandemic [17]. Indeed, studies during the COVID-19 pandemic, such as those by Blanco-Donoso, et al. [17] and Sarabia-Cobo et al. [18], have noted that despite significant stress and emotional exhaustion, many caregivers in nursing homes reported a high level of professional satisfaction and a robust sense of duty and commitment. It is crucial to acknowledge that these do not necessarily eliminate the challenges or the emotional burden associated with caregiving. Recognizing this coexistence helps to avoid romanticizing caregiving and highlights the complexity of the caregiving experience, particularly in crisis situations.

Given this context, the aim of this study is to explore the positive outcomes experienced by caregivers in nursing homes during the COVID-19 pandemic. Through qualitative research, we sought to understand how these caregivers—spanning a range of roles including nurses, medical staff, psychologists, physiotherapists, and support personnel—experienced personal and professional growth despite the immense challenges. This study focuses on the positive outcomes to address a less-explored aspect of the caregiving experience; however, it is important to note that this approach does not imply ignoring the significant challenges and negative impacts faced by caregivers. The insights from this study can inform strategies to support caregivers and guide the development of systems that balance infection control measures with holistic resident care.

## Methods

### Design

This exploratory study uses qualitative methodology with an interpretive approach to gain a comprehensive understanding of participants' experiences. The decision to employ a qualitative methodology was driven by two primary factors. Firstly, qualitative methods facilitate a deeper understanding and provide a richer, more detailed description compared to quantitative methods [19]. Secondly, the chosen design aimed at contrasting various professional viewpoints through meticulous analysis of the data [20].

### Study participants and recruitment

The study focused on Spanish nursing home professionals who had worked in the facility from the first wave of the COVID-19 pandemic until the time of the study.

There were no exclusion criteria. Two nursing homes participated in the study, and all professionals who wished to participate were interviewed. The absence of exclusion criteria was intentional to ensure a comprehensive understanding of the diverse experiences and perspectives of all professional caregivers in the nursing homes. By including all willing participants, the study aimed to capture a broad range of insights and adaptations, providing a more holistic view of the positive outcomes and challenges faced during the COVID-19 pandemic. This approach allowed for a richer and more inclusive analysis, reflecting the varied roles and experiences of the entire caregiving team.

### Interviews

We aimed to conduct interviews with a diverse sample of professional caregivers in nursing homes to explore their experiences during the COVID-19 pandemic. In order to capture a wide range of care experiences, we selected both a public and a private nursing home. This approach aimed to avoid potential biases related to the management or ownership of the facilities, as these factors may influence operational practices and care dynamics. However, the intention was not to carry out a comparative analysis between public and private settings, but rather to enrich the diversity of perspectives in the study.

A wide range of care roles were also included to gain a holistic understanding of the care experience in care homes. By interviewing people from different professional backgrounds - including medical staff, care assistants, therapists, cleaners and administrators - we aimed to reflect the interconnected and collaborative nature of care in these settings. This approach allowed us to gain insights into the unique contributions and challenges of each role, enriching our understanding of the shared and distinct experiences of the entire care team during the COVID-19 pandemic.

A total of 15 interviews were planned, based on our intention to obtain a comprehensive, exploratory understanding of the positive and negative outcomes experienced by caregivers during this period. The interview guide [21] consisted of 17 core questions: five focused on the work context and roles, five on COVID-19 experiences, one on negative aspects, and seven on positive outcomes. The questions were initially reviewed by a geriatric physician with experience in nursing

home settings to ensure clarity and relevance within this professional context. Additionally, we conducted a cultural adaptation through a pilot interview, which allowed us to confirm that the questions were clearly understood, culturally appropriate, and effectively framed for this specific setting [22,23]. This step enabled us to make any necessary adjustments to improve the guide's clarity and ensure alignment with the study's objectives.

The semi-structured interview script was developed by a team of three clinical psychologists (see S1 File): one senior researcher (M.S.I., Ph.D.) with expertise in gerontology and two senior researchers (M.P., Ph.D., and A.O., Ph.D.) with a background in qualitative research. This guide was designed to ensure consistency across interviews and to facilitate a rich exploration of caregivers' experiences, while allowing for the flexibility needed to probe individual perspectives in more depth.

The recruitment process began with the directors of nursing homes, who informed caregiving staff about the study and provided information about voluntary participation. Caregivers who expressed interest shared their contact details, and the research team reached out by telephone to confirm participation, discuss preferred interview format, and schedule a convenient time.

From December 1st, 2021 to July 5th, 2022, interviews were conducted either online or by telephone, depending on participant preference. Participants confirmed their agreement to participate and for the interview to be recorded immediately prior to the start of each session. They were provided with detailed information about the study's aims, procedures, and their rights, including the voluntary nature of their participation and their right to withdraw at any time without any negative consequences. Interviews lasted between 16 and 53.14 minutes, with an average duration of 36.8 minutes. For online interviews, conducted via Microsoft Teams, the platform's recording function was used; for telephone interviews, a tablet device was used to record the audio. They were transcribed by the research team without participant review. To protect confidentiality, all recordings were securely stored and transcriptions were anonymized, with identifying information removed during transcription.

Two researchers (M.S.I. and M.P.) and a male master's student in clinical psychology (R.G.) conducted the interviews. There was no prior relationship between the interviewers and participants, except for the initial recruitment contact. Participants were informed of the interviewers' professional roles and affiliations, and the study's purpose and motivations were explained before each session. Additionally, participants were assured that their responses would remain confidential and that any published results would be aggregated to prevent identification.

Interviews were conducted either via telephone (13 participants) or videoconference (2 participants), according to participant preference. Three individuals declined to participate, citing discomfort with discussing the topic. The study received ethical approval from blinded for review (Approval No.2021/83 dated June 21st, 2021), and informed consent was obtained from all participants prior to the interviews.

## Data analysis

Coding and data analysis were supported with the software ATLAS.ti Scientific Software Development GmbH (https://atlasti.com). A qualitative content analysis method was used to perform data analysis with a combined inductive and deductive coding frame. The objective was to identify specific themes and concepts, while also accommodating ideas that were not originally derived from theoretical frameworks [24]. The analysis began with a preliminary code list derived from previous literature and adhered to the guidelines established by Braun and Clarke [25] to identify themes related to COVID-19 experiences and aspects of caregiving.

The categorization process followed in the study is visually represented in Fig 1, which summarizes the steps from initial coding to theme development.

Three researchers (M.S.I., M.P., A.O.) were responsible for coding the transcripts. Initially, they utilized a preliminary list of codes, but as the interviews progressed, new codes naturally emerged. To manage discrepancies, each researcher independently coded one transcript and then compared their coding decisions. Discrepancies were discussed in regular

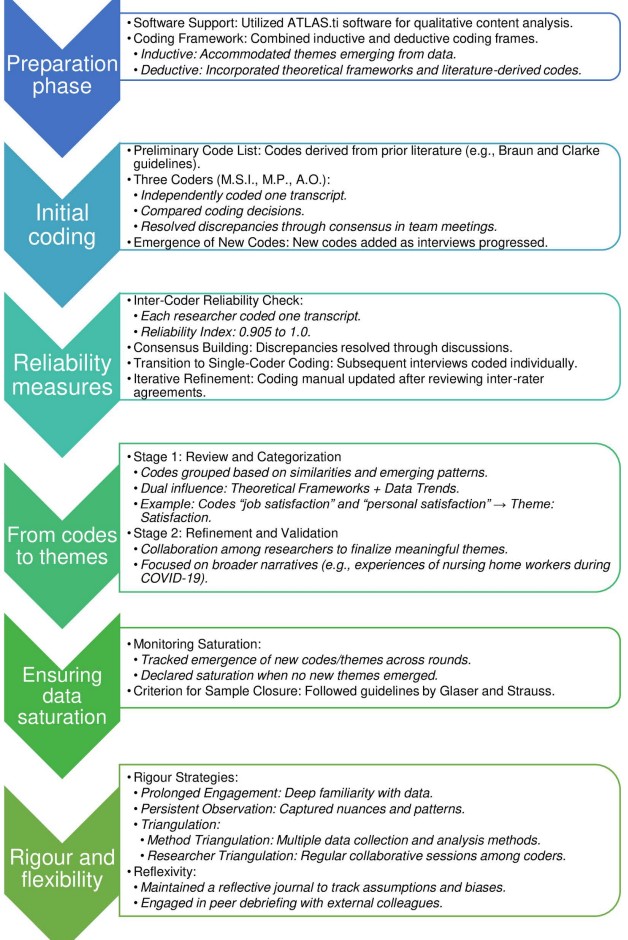

**Fig 1. Coding Categorization Diagram.**

meetings until a consensus was reached, ensuring consistency and reliability in the coding process. The coding manual, which was iteratively refined during this phase, is detailed in S2 Table.

To ensure the rigor of the study, several strategies beyond inter-coder reliability were implemented. These strategies encompassed prolonged engagement with the data, persistent observation, and the triangulation of researchers and methods.

From identified codes to themes involved reviewing and categorizing codes based on observed similarities and patterns. For instance, codes pertaining to "job satisfaction" and "personal satisfaction" were amalgamated under the comprehensive theme of 'Satisfaction'. In a similar vein, codes like "cohesive team", "resilience", and "social support" were grouped under the theme "Positive Aspects of Caregiving". Fig 1 provides a visual summary of this process.

In terms of data saturation, the approach was continuously evaluated throughout the analysis process. Data saturation was considered to have been reached when no new themes or insights emerged from additional interviews. This was monitored by tracking the emergence of new codes and themes during each round of coding. By the final interviews it was observed that no new themes were emerging, suggesting that data saturation had been reached. In line with the qualitative methodology outlined by Glaser and Strauss [26], saturation functioned as our criterion for sample closure, ensuring a thorough exploration of the phenomenon under study.

### Rigor and reflexivity

Ensuring the rigour and reflexivity of the study was a fundamental aspect of the research process. Several methodological strategies were employed to maintain high standards of rigour. These included prolonged engagement with the data, persistent observation and triangulation of data sources, researchers and methods. Each of these strategies was implemented to enhance the credibility, transferability, dependability and replicability of the findings.

Prolonged engagement involved spending considerable time with the data to fully understand the context and nuances of carers' experiences. Persistent observation allowed the researchers to identify patterns and themes that emerged from the data, ensuring that the analysis was grounded in the participants' experiences.

Triangulation was achieved by involving multiple researchers in the data analysis process, which helped to minimise individual bias and provide a comprehensive understanding of the data. Method triangulation involved the use of multiple methods to collect and analyse the data, which increased the validity of the study. Researcher triangulation involved regular discussions and debriefings among the research team to compare and contrast different perspectives, ensuring a robust analysis.

Reflexivity was also an important aspect of the research process. The researchers kept a reflective journal throughout the study to document their thoughts, assumptions and potential biases. This practice helped the researchers to remain aware of their influence on the research process and findings. By continually reflecting on their positionality and its impact on the interpretation of the data, the researchers aimed to maintain the integrity and authenticity of the study.

In addition, researchers engaged in peer debriefing sessions with colleagues not directly involved in the study. These sessions provided an external check on the research process and helped to identify any blind spots or biases that might have influenced the analysis. Member checking was not carried out in this study due to logistical constraints, but the rigorous approach taken by the research team aimed to mitigate this limitation.

To ensure transparency in reporting the research, the Consolidated Criteria for Reporting Qualitative Research [27] was used (see S3 File).

## Results

### Participants characteristics

A total of 15 workers in different positions in two nursing homes (one public and one private) participated in the study. The participants' main characteristics are detailed in Table 1. The mean duration of working with older adults was 10.9±7.9 years (range 2–30 years), and time working in the current nursing home was 8.8±7.0. Nearly two-thirds of the participants were female (73.3%), and the remaining 26.7% were male.

### Major themes

From the interviews, we identified 15 positive themes about the nursing home workers' experiences during the COVID-19 pandemic, as detailed in Table 2. We'll discuss these categories further using quotes from participants, following Sandelowski's guidelines [28] for illustrating themes.

In the following sections, we'll focus on issues that were emphasized by more than half of the participants. Following qualitative research principles, we included themes reported by at least 40% of participants, as these represent shared experiences that add depth and coherence to the analysis. Topics mentioned by fewer participants were considered non-saturating and thus were not included in the main findings, aligning with the "winnowing" approach", which emphasizes distilling the data to central, representative themes that illuminate key aspects of participants' experiences [23]. Relevant participant quotes, properly identified and cited throughout the document, were included in the manuscript to illustrate findings and themes.

**Table 1. Characteristics of the participants.**

| Professional Role | Sex | Years working with older adults | Years working in the current nursing home |
|---|---|---|---|
| Director | Female | 12 | 2 |
| Doctor | Male | 3 | 2 |
| Doctor | Male | 8 | 8 |
| Social worker | Male | 4 | 4 |
| Physiotherapist | Female | 3 | 2.5 |
| Receptionist | Female | 15 | 15 |
| Receptionist | Female | 7 | 7 |
| Caregiver Coordinator | Female | 20 | 20 |
| Geriatric Aide | Female | 11 | 2 |
| Geriatric Aide | Female | 3 | 3 |
| Nursing Assistant | Male | 11 | 11 |
| Nursing Assistant | Female | 30 | 18 |
| Nursing Assistant | Female | 17 | 17 |
| Hairdresser | Female | 18 | 18 |
| Cleaner | Female | 2 | 2 |

**Table 2. Positive aspects related to the experience of COVID-19 in nursing homes.**

| | N |
|---|---|
| Post-traumatic growth | 14 |
| Social support | 14 |
| Cohesive team | 13 |
| Improvement in relationships: family relationship (n=6) and with the residents' families (n=6) | 12 |
| Resilience | 12 |
| Finding meaning | 11 |
| Satisfaction: job satisfaction (n=11) and personal satisfaction (n=8) | 11 |
| Gratitude (n=8) and grateful response | 7 |
| Humanization | 6 |
| Joy | 5 |
| Self-efficacy | 5 |
| Communication | 5 |
| Faith/religión | 5 |
| Empathy | 1 |
| Acceptance | 1 |

## Post-traumatic growth

Among the fourteen participants who reported positive mental shifts due to navigating challenging situations, eleven specifically noted enhancements in their problem-solving, coping, and organizational abilities. These skills were instrumental in managing their tasks effectively without becoming overwhelmed. For 11 interviewees, these changes included a reevaluation of priorities, which included recognizing and appreciating the value of life and important issues. For example: "*After this, uh... you realize how important life is [...] to care about what really matters*" (P5).

Six participants emphasized that their experiences had fostered greater optimism and a more positive outlook on life. For example: "*I think sometimes I could be working, and the day could be a fatal day that things go wrong, but I have this thing of saying: "this is small, this doesn't have...this is solved" and being more optimistic, I think so. I'm more optimistic every day, in everything, in everything*" (P3).

Another participant shared a significant shift towards optimism and improved organizational skills:

"*I am more optimistic, and it has given me more capacity to work [...] I have the capacity to absorb more problems. Maybe...I don't know, before there were a hundred problems in one minute and, now, I don't know, if ten problems come in one minute, I know how to organize them, I know how to organize them better. I am more optimistic and I know how to organize problems better, I mean, after a moment when everything was disorganized and you did not even know what you were going to do, it seems to me that now I am better organized in the day to day work, in what are the priorities, what is important and what is less important and I feel more optimistic in everything in life, in life and in work and I did not lose sleep [laughs], you know that this is an important positive thing*" (P2).

Importantly, this optimism was not characterized as a denial of distressing moments but rather as a capacity to integrate such challenges into a broader perspective, allowing for constructive action despite adversity. This aligns with frameworks suggesting that optimism involves a balanced cognitive process that acknowledges distress while focusing on solutions and opportunities [29,30], a process that has proven particularly relevant during the COVID-19 crisis [31].

Two interviewees expressed that accompanying the families of deceased residents was a crucial aspect of their experience. They emphasized that maintaining communication with families, updating them about their loved ones, and ensuring they were informed, was not only greatly appreciated by the families but also became a source of personal fulfillment for the caregivers. This aspect of their role, they reflected, helped them to cope more effectively with the challenges they faced and to discern positive elements amidst their experiences:

"*With the families [...] calling them, telling them how they were... I don't know, I think that's what I mostly take and I think that's what the families appreciated the most and the people were grateful. In the end, you called them, you told them how they were... they knew that you had seen them that day... they knew that you had been with them... I don't know, because that was a little bit... what I take, I mean, because it's true that later on that's what they appreciated the most*" (P2).

### Social support

The professionals emphasized the vital role of community support in navigating the challenges of the pandemic. This support, which came from neighbors, peers, friends, and family, was essential to coping with the whole experience. Illustrating this, one participant recounted the meaningful gestures of support from neighbors, such as providing food, handmade protective equipment, and other necessities: "*The neighbors brought us food, they brought us dinner, they gave us garbage bags with sleeves, handmade masks... in other words, the truth is that, in this sense, these are positive things that in the end, people turned to us*" (P13).

Family and friends' support was underscored as a critical element in the resilience of healthcare professionals during the pandemic. An interviewee shared an example of this support, recounting conversations with her mother, who provided emotional encouragement from a distance due to safety precautions: "*without the support, it must be a lot harder, right? I remember chatting, sitting in my room, with everything open and my mother in the hallway and, "How was your day?" or "Yes, how was your night?" And, whatever, and "come on, my child, you can do it, this is hard, but in the end*…" (P13).

The broader societal support played a significant role in uplifting the spirits of healthcare professionals. One interviewee highlighted the daily presence and contributions of municipal police officers and members of the local community,

including neighborhood associations: "*the municipal police officers were there every day. They always brought us "little things", and people from the street, from the neighborhood association... they bought us things too*" (P7).

Peer support also emerged as a prominent theme consistently highlighted by nearly all participants (n=14), enabling them to navigate the experience while fostering a sense of cohesion. One participant reflected on the initial overwhelming nature of the experience and the crucial support provided by coworkers: "*the truth is that at the beginning it was a bit overwhelming and at least we had the colleagues, especially in the nursing home, the people you saw every day were the ones who helped you*" (P2). Another participant described the development of deep friendships akin to a family bond, where mutual support was a constant: "*The unconditional support of our colleagues, we have really also formed a very nice friendship, just like a family, we basically support each other, whatever happens, we're there and that's what it's all about, it's fine. We also had to share special moments*" (P11).

The participants in the study expressed gratitude for the support they received, both emotional and instrumental. Six participants specifically mentioned the emotional support from colleagues, family, and friends, often exemplified by caring inquiries over the phone about their well-being: "*From colleagues, from my family...yes...that's right. Also from friends on the phone, from a phone call:* "Hello, how are you? how are you doing?""(P8). In contrast, eight participants highlighted the practical or instrumental support that significantly eased their daily lives outside work. One participant, for example, shared how her sibling provided meals, a gesture that alleviated the stress of having to cook after long work hours and was greatly appreciated for its practical help during a demanding time:

> "For example, my sister lives very close to me, we live very close to each other, and then she said to me: "Hey, I have to cook for four, I can cook for four or five, I'll put it in the tupperware, I'll send it to you in the elevator and you eat it". So... I, well, see, it is true that I was very grateful, because for me to come home at 3:30 pm and think that I have to start cooking or something, well...I didn't feel like it. Probably I wouldn't have had anything to eat or I would have eaten badly, so it was a huge support for me" (P9).

## Cohesive team

Most interviewees (n=13) highlighted the emergence of a cohesive team at their workplace during the pandemic. This team was characterized by a deep sense of unity, collaboration, and mutual support, creating a respectful and supportive work environment. Within this environment, team members demonstrated mutual trust, actively shared information, and collaborated effectively to meet the complex needs of residents. One participant vividly described this environment, praising the extraordinary team effort, the absence of complaints despite challenging circumstances, and the unparalleled level of teamwork experienced during this period: "*For me it was impressive to see how everyone worked their butts off. Nobody complained, and we were all a mess, and the teamwork was brutal. I mean, I had never worked as a team like that before, you know? I mean, side by side, everybody was there... for me that was the best part of it*" (P9).

This experience has also brought a deep sense of gratitude for their colleagues and the invaluable support they have given each other. They reflected on the invaluable support and camaraderie shared during the pandemic, which made navigating the challenges more manageable. A participant really appreciated the team, expressing heartfelt thanks to them, acknowledging how much easier things were because they had a great group of assistants, nurses, and doctors around them:

> "*Well, when things happen, we laugh and look back and...and tell them that...if I had to tell them something I would say thank you very much for being there with me in those moments, because in the end it has been easier for me having the great team I have at work, right? Because I can't say anything else. Whether they are assistants, nurses, or doctors, all of them have proved to be the best*" (P13).

### Resilience

Twelve participants shared their experiences of resilience in coping with the stress and demands of their profession during the pandemic. These narratives highlighted not just coping but emerging stronger from the experience. One participant described it as an unexpected "war," emphasizing their growth in strength and better organization as a result: *"A war that we were not expecting, that came in days, and we were not prepared mentally or logistically, but well, we came out of it. I think we came out stronger and better organized"* (P3). Another participant echoed this sentiment, stating a significant increase in their psychological strength: *"I feel even stronger, much stronger psychologically"* (P6).

Several caregivers identified key moments when they experienced a shift in mindset that allowed them to effectively manage the situation, recover, and approach their caregiving responsibilities with a renewed sense of purpose and satisfaction. For instance, one caregiver noted:

"*I think I got to a point where I said: "This is not going finish me off, this has to change". Then is where I said: "you are part of something, you have to move forward" and, well, I think that saved me a lot and I moved on (...) I saw that I was willing to work, willing to help, willing to fight... and they were there waiting to see what would happen, right? And being here helped me a lot*" (P14).

### Finding meaning

Eleven participants expressed positive feelings about caregiving, describing it as an educational and rewarding experience and recognizing the strengths inherent in their role. They recognized the inherent strengths in their profession. For example, on individual spoke of the small, joyful interactions with residents and the mutual benefit of this relationship, highlighting how caregiving inspired feelings of tenderness and affection:

"*My attitude changed...I began to see little things, [...] I noticed that [...] the little time I was with them, I was happier, and I noticed that I was helping myself by helping them, so I said: "this is the way" [...]. I tell the residents that it is not me that helps them, it's them who help me because they inspire in me so much tenderness and so much affection that I say "thank you," because I don't want them to think that I'm doing them a favor, they're doing me a favor because they bring out the best in me. Even the most fussy, even the most fussy that is there, because after a moment you win them and they are a love*" (P14).

### Satisfaction

Job satisfaction was a topic discussed by eleven interviewees who expressed that they like their jobs and are satisfied with them. For example, one participant enthusiastically affirmed his love for the job: "*Yes, it is a job that I like, that I like a lot*" (P10).

Additionally, eight participants highlighted the personal satisfaction derived from providing the best possible care to another person. One participant, for instance, shared a sense of pride and happiness in her contributions, noting the lasting impact on both residents and the appreciation received from colleagues: "*I have the feeling of...of being happy with what I have done. Of having helped many who are still with us and are still with us and the colleagues' affection, you know? I am proud of that*" (P8).

While participants expressed satisfaction with their actions and contributions, it is important to acknowledge that this sense of fulfillment coexisted with significant emotional and physical demands. Caregiving during the pandemic presented immense challenges, and feelings of pride and accomplishment did not negate the ongoing stress and

burdens faced by caregivers [32]. For instance, one participant voiced a strong sense of pride in their accomplishments and an unequivocal willingness to take the same actions again if faced with a similar situation, even while acknowledging the effort required:

> *"I feel very proud of what I have done so far, what I have tried to help in whatever way I could...and if it happens again, I would do the same thing again, you know? I have no doubt about that (....) Satisfied, satisfied with what...more could have been done, but well...satisfied with what has been done...and we are still doing...and we are still doing"* (P8).

These findings underscore the dual nature of caregivers' experiences, where professional satisfaction and pride coexist with the emotional and physical tolls of caregiving, reflecting the complex and multifaceted reality of their roles.

They not only expressed satisfaction with their actions and contributions but also a readiness to respond similarly in the future. For instance, one participant voiced a strong sense of pride in their accomplishments and an unequivocal willingness to take the same actions again if faced with a similar situation:

> *"I feel very proud of what I have done so far, what I have tried to help in whatever way I could...and if it happens again, I would do the same thing again, you know? I have no doubt about that (....) Satisfied, satisfied with what...more could have been done, but well...satisfied with what has been done...and we are still doing...and we are still doing"* (P8).

## Gratitude

The sentiments of gratitude expressed by various participants in the study reflect a multifaceted appreciation for the resilience, cooperation, and positivity encountered during the pandemic. Four participants expressed their gratitude to the residents for their resilience and strength during this challenging experience: "*I would thank the residents for their cooperation in this situation when they were alone and you came in and told them that they had to stay there alone, that there were a few days left, but we didn't know how many and....I would thank them for that, for the smiles they gave you when you arrived in the mornings, you know? And they saw you*" (P10).

The professional caregivers appreciate the positive and grateful response of the residents to the care they provided (n=5). They emphasize that even when verbal communication is limited, the older people express their gratitude through nonverbal communication: "*Look, for a smile [...] we have a man in the nursing home who doesn't speak or anything, right, and every day his children would take him out. Of course, with the pandemic, we were all closed...The first day they brought him to the reception, he raised his arms and gave us a smile!*" (P7).

Additionally, six individuals expressed their appreciation to their colleagues for the invaluable support they provided each other during this time: "*I would also thank my colleagues for having worked with everyone, I mean, unconditionally with everyone. Yes, that is what I would do, I mean, everybody..., I would thank everybody for the work they have done, because... at least in the nursing homes where I worked, I think that...I mean, that was wonderful, the job done by everyone*" (P2).

Furthermore, four interviewees expressed their gratitude to the supportive individuals who showed genuine concern for their well-being. For example: "Even though we were in that area and we couldn't leave until we went [home], that little call of "*how are you? how are you doing?*", that's appreciated, isn't it? Because they care about you, and we have been there for many years" (P13).

Four employees appreciated the expressions of gratitude from the resident's family members, who expressed their appreciation for the employees' dedication and compassionate care during the difficult times of the pandemic. For example: *"here, for example, relatives have thanked us many times [...] They brought us cakes, pastries...those that have seen the work we have done, they have thanked us*" (P14).

### Improvement in relationships

Half of the participants noted improved relationships with both their own families and the families of residents following the pandemic. Six specifically mentioned a positive improvement in their family relationships. For instance: "*How we did it with the families...that was very important, and it had a big impact on me, I think it taught me a lot. Thank God nobody in my family had a bad time because of it, so...I lived it in a way that was very nice in that sense, you know? From...hey, my family has moved forward, they have been solid*" (P4).

Four professionals (26.7%), all in roles that involve direct interaction with families, expressed satisfaction with these relationships, prompting comments from two of them: "*Hey, we bring you...*", in the beginning they brought us masks when there were no masks: "*I bring you a tray with cakes*" (P4). Another person said: "*I leave sweet fritters at the reception for you to have breakfast...*". *So, they saw that we cared for their relatives, that maybe they did not see their relatives and we did, and even when they came back they [the old people] would not go with them [their relatives], they came with us because we had been their reference persons for three months, when they had not seen them*" (P7).

## Discussion

This study analyzes the experiences of formal caregivers in nursing homes during the COVID-19 pandemic, shedding new light on caregiving during the COVID-19 crisis in Spain. Our findings reveal a complex picture of caregiving in Spain during the pandemic. While previous studies have highlighted similar challenges faced by caregivers in other countries (e.g., [7,33] and in Spain [18,34], our study contributes unique insights into the specific conditions and responses within Spanish nursing homes. Distinctively, our research emphasizes the positive adaptations and growth experienced by caregivers, highlighting resilience and growth as crucial outcomes often overlooked in pandemic-related literature.

The pandemic exacerbated challenges by increasing workloads and emotional pressures (e.g., [2,34,35]). Caregivers faced additional tasks and responsibilities, uncertainty, and grief over resident losses, mirroring broader research on pandemic caregiver stress (e.g., [7,36]), including Spanish nursing homes [34,37]. According to Tedeschi and Calhoun [29], growth cannot occur without an initial psychological disruption that challenges pre-existing beliefs and creates the need to rebuild a sense of coherence. This process does not eliminate pain but enables individuals to find meaning and purpose even while experiencing emotions such as sadness, loss, or frustration. In this sense, post-traumatic growth does not replace or negate the negative effects of trauma; rather, both elements coexist in a dynamic balance. Individuals can experience resilience and growth while simultaneously facing symptoms such as anxiety or stress [32]. Our findings highlight how crisis conditions fostered significant post-traumatic growth and professional enrichment among carers, with many developing enhanced problem-solving skills, reappraised life priorities, and found meaning in their situation. They expressed satisfaction with the work carried out, always seeking the welfare of the elderly, and gratitude to the residents for their positive response to their care, to their colleagues for their support and to all those who supported them during this difficult situation.

Protective measures while safeguarding the elderly inadvertently increased isolation. These risks presented considerable challenges for formal workers, who had to acknowledge their own risk to older adults and ensure transparency among them to reduce dangers [38]. As challenging as it was, this separation also led some carers to develop stronger bonds and deeper professional commitment, increasing their resilience and ability to cope with professional demands, thus turning a potentially negative situation into an opportunity to strengthen team dynamics and personal growth. Previous studies have shown the resilience that characterizes LTC staff [39].

These findings emphasize the impact of the pandemic and the urgent need for comprehensive support, consistent with global sentiment [2,12]. Amidst the challenges of the pandemic, our study uncovered positive outcomes as caregivers displayed post-traumatic growth, enhanced team cohesion, and a renewed sense of professional purpose. These positive outcomes did not emerge in isolation from significant emotional burdens; the duality of caregiving experiences—where

positive adaptations coexist with profound emotional distress—reflects the complex and multifaceted nature of caregiving during crises. This positive adaptation aligns with research on healthcare workers [13,40], suggesting that the crisis context, while challenging, can also serve as a catalyst for significant personal and professional development. In particular, developing a heightened ability to navigate challenges strengthened self-discipline and a greater appreciation for life, which resulted in significant positive changes, pivotal in contexts of ongoing healthcare challenges.

The protective role of social support, as emphasized by our participants, echoes findings across pandemic research [34]. The pandemic strengthened team cohesion, with participants reporting deepened team bonds that improved care and emotional support [41]. Moreover, the emergence of a bottom-up management approach in some settings [42] underscores how crisis-driven conditions can catalyze adaptive organizational changes, fostering a participatory management style that empowers caregivers, which not only engendered a sense of utility among employees but also endowed them with the confidence and autonomy necessary to perform their duties in extremely challenging, crisis-driven conditions.

A 2020 survey highlighted caregivers' renewed professional pride and accomplishment, factors associated with job satisfaction and engagement [43]. Cohesive teams, as observed, can alleviate distress through mutual support [44].

Our data underscore the resilience exhibited by caregivers, aligning with prior research that has shown healthcare workers maintained resilience during the COVID-19 pandemic [45,46]. However, it is crucial to acknowledge that resilience does not negate the emotional burdens of caregiving; instead, it reflects the ability to navigate these burdens while maintaining a sense of purpose and functionality. Resilience plays a critical role in both personal and professional domains, serving as a protective factor against the adverse effects of workplace stress. The co-occurrence of bereavement and resilience among participants is noteworthy, suggesting coping and adaptability [38,47]. This finding underscores the complexity of resilience, emphasizing that it is not synonymous with the absence of hardship but rather a dynamic process that integrates emotional challenges with the capacity to recover and grow. A study conducted in hospitals designated for COVID-19 treatment in Korea indicated that nurses' resilience not only directly contributed to enhanced post-traumatic growth but also exerted an indirect effect by fostering a sense of meaning in life [48]. The findings suggest that COVID-19 may act as a catalyst for resilience, offering a shield against stress and subsequently fostering post-traumatic growth [49]. Furthermore, there is evidence that resilience is a predictor of finding meaning in life [50].

Lastly, participants derived satisfaction from their roles, which is echoed in healthcare studies [51]. Our findings correspond with a study conducted in Spain, which indicated that despite facing substantial job demands, limited job resources, the fear of contagion, and exhaustion, nursing home workers in Spain reported high levels of satisfaction during the COVID-19 crisis [17]. Their essential roles for the elderly reinforced the profound impact of their work and increased job satisfaction, highlighting the critical importance of recognizing and supporting the emotional and professional needs of caregivers during and after health crises.

It is important to emphasize the distinction between the focus on the positive aspects identified in our study and toxic positivity or the trivialization of suffering. For post-traumatic growth to occur following adverse events, new trauma-related information can only be processed in two ways: it is either assimilated into existing world models (which would imply a lack of elaboration necessary for growth) or accommodated within those models. To transcend the pre-trauma baseline, accommodation rather than assimilation is required, as growth, by definition, involves the development of new worldviews and a significant effort in the elaboration of the traumatic experience. While the alleviation of distress can result from either assimilation or accommodation processes, only accommodation fosters growth. In contrast, assimilation may leave an individual more vulnerable to future retraumatization.

This study has several limitations. First, the use of a convenience sample limits generalizability to broader populations but is consistent with the goal of examining contemporary changes in caregiving during a crisis, justifying this sampling method. Second, potential biases in caregivers' responses, including recall bias, social desirability, or researcher presence, may affect the authenticity of their accounts. Despite these limitations, the study provides a rich, in-depth

exploration of caregivers' experiences and offers valuable insights that contribute to understanding resilience and post-traumatic growth in high-stress environments.

Our study contributes valuable insights into the resilience and growth of caregivers during the COVID-19 pandemic, illustrating how positive dynamics can emerge in high-stress environments. This study is essential for organizations operating in the nursing home sector because it provides an in-depth understanding of the impact of the COVID-19 pandemic on nursing and allied health personnel. This is vital to facilitate staff preparedness for future adversity and to enhance organizational resilience. Professionals working in these settings have consistently proved a commendable ability to achieve positive outcomes in the midst of the challenging circumstances of the pandemic. This resilience and post-traumatic growth are critical elements that must be incorporated into future interventions aimed at enhancing their overall well-being and effectiveness.

At this point, it is essential to acknowledge the persistent emotional burdens that coexist with these positive dynamics, ensuring that interventions provide holistic support that addresses and validates the challenges caregivers face. The process of accommodation described earlier requires significant cognitive effort, and any intervention that impedes this process may prove detrimental. Furthermore, adversity does not inherently lead to positive change for everyone, and it is crucial to avoid conveying that there is something inherently positive in every traumatic experience. The pressure to "overcome" trauma quickly can create feelings of isolation for those who do not experience immediate or visible growth; as Calhoun and Tedeschi (1998) [52] stated: "...it is important to use language that clearly situates the impetus for growth within the struggle against the event, not in the event itself" (p. 366).

Further research should aim to conduct longitudinal studies to track the long-term effects of the COVID-19 pandemic on carers' mental health, resilience, and professional development. Increasing the size and diversity of the sample to include a wider range of care homes in different regions and settings will improve the generalizability of the findings. It is also important to investigate specific factors that contribute to resilience and post-traumatic growth, such as individual coping strategies, organizational support mechanisms, and community resources. In addition, exploring the role of training and professional development in fostering resilience among carers, with a particular focus on interventions that can be delivered in crisis situations, will provide valuable insights. Finally, exploring the impact of policy changes on the well-being of carers, and assessing how different support systems and protections influence their experiences and outcomes, will help inform future strategies to support this vital workforce.

## Conclusions

The COVID-19 pandemic highlighted the dual experience of nursing home caregivers in Spain. While they faced significant emotional distress and overload, they also displayed remarkable resilience and post-traumatic growth. Such adaptability not only underscores their unwavering commitment in the face of adversity but also highlights the crucial role caregivers play in healthcare systems. This research emphasizes the profound dedication and commitment of nursing staff and other personnel in senior living facilities during the pandemic, as they navigated a plethora of challenges to deliver optimal care to residents.

The positive outcomes observed—such as enhanced problem-solving abilities, post- traumatic growth, resilience, stronger team cohesion, and finding meaning and satisfaction—demonstrate how crisis conditions can catalyze substantial personal and professional development among caregivers. Future studies should continue to explore these positive outcomes, aiming to develop strategies that not only mitigate the negative impacts of such crises but also promote and sustain the beneficial changes. A particular focus should be placed on enhancing team dynamics and support systems in nursing homes to bolster resilience and improve care outcomes.

As we emerge from the pandemic, it is vital to prioritize comprehensive support for carers. This support should cover both the emotional and professional aspects of their role, ensuring that carers are equipped to manage stress and maintain their wellbeing. Adequate support for carers is essential not only for their own health, but also for the well-being of the

older people they care for. Investing in the resilience and growth of caregivers not only prepares them for future challenges, but also improves the overall effectiveness of care in retirement communities.

## Supporting information

**S1 File. Semi-structured interview.**
(DOCX)

**S2 Table. Coding manual.**
(DOCX)

**S3 File. COREQ.**
(PDF)

## Acknowledgments

We extend our gratitude to the nursing home staff for their participation in this study and their significant work throughout the pandemic.

## Author contributions

**Conceptualization:** Macarena Sánchez-Izquierdo, María Prieto-Ursúa, Ángela Ordóñez-Carabaño.

**Data curation:** Macarena Sánchez-Izquierdo, María Prieto-Ursúa.

**Formal analysis:** Macarena Sánchez-Izquierdo, Rodrigo García-Sánchez, María Prieto-Ursúa, Ángela Ordóñez-Carabaño.

**Funding acquisition:** Macarena Sánchez-Izquierdo.

**Investigation:** Macarena Sánchez-Izquierdo, Rodrigo García-Sánchez, María Prieto-Ursúa, Ángela Ordóñez-Carabaño.

**Methodology:** Macarena Sánchez-Izquierdo, María Prieto-Ursúa, Ángela Ordóñez-Carabaño.

**Project administration:** Macarena Sánchez-Izquierdo.

**Resources:** Macarena Sánchez-Izquierdo, María Prieto-Ursúa, Ángela Ordóñez-Carabaño.

**Software:** Macarena Sánchez-Izquierdo, María Prieto-Ursúa.

**Supervision:** Macarena Sánchez-Izquierdo, Jesús Mateos-Nozal.

**Validation:** Macarena Sánchez-Izquierdo.

**Visualization:** Macarena Sánchez-Izquierdo.

**Writing – original draft:** Macarena Sánchez-Izquierdo.

**Writing – review & editing:** Macarena Sánchez-Izquierdo, María Prieto-Ursúa, Jesús Mateos-Nozal, Ángela Ordóñez-Carabaño.

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
