## [Decision Letter · Decision Letter 0]

7 Nov 2024

PONE-D-24-43127Positive outcomes among nursing home caregivers in Spain during the COVID-19 pandemic: A qualitative interview studyPLOS ONE

Dear Dr. Ordóñez-Carabaño,

Thank you for submitting your manuscript to PLOS ONE. After careful consideration, we feel that it has merit but does not fully meet PLOS ONE’s publication criteria as it currently stands. Therefore, we invite you to submit a revised version of the manuscript that addresses the points raised during the review process.

We look forward to receiving your revised manuscript.

Kind regards,

Anna Rachel Conolly, PhD, MSc, PG Dip, BA (hons)

Academic Editor

PLOS ONE

**Journal Requirements:**

This work was supported by Comillas Pontifical University through the 2022 call for internal research projects. The research project is entitled “Positive Effects of the COVID Pandemic on Professional Caregivers of Elderly Individuals in Nursing Homes”.

4. In the online submission form you indicate that your data is not available for proprietary reasons and have provided a contact point for accessing this data. Please note that your current contact point is a co-author on this manuscript. According to our Data Policy, the contact point must not be an author on the manuscript and must be an institutional contact, ideally not an individual. Please revise your data statement to a non-author institutional point of contact, such as a data access or ethics committee, and send this to us via return email. Please also include contact information for the third party organization, and please include the full citation of where the data can be found.

5. Please amend your list of authors on the manuscript to ensure that each author is linked to an affiliation. Authors’ affiliations should reflect the institution where the work was done (if authors moved subsequently, you can also list the new affiliation stating “current affiliation:….” as necessary).

**Additional Editor Comments:**

Dear Authors

I agree with both reviewer's comments and recommend a major revision for your paper. The transparency and methodological robustness will be increased by paying close attention to the suggested revisions such as providing greater clarity regarding how participants were sampled and recruited. I would also value a discussion of the themes that you did not include - those that were reported by less than 40% of the sample. 40% of the population is still a significant number. I look forward to reading your revised paper.

Best wishes

Anna Conolly

Reviewers' comments:

Reviewer's Responses to Questions

**Comments to the Author**

1. Is the manuscript technically sound, and do the data support the conclusions?

Reviewer #1: Partly

Reviewer #2: Yes

2. Has the statistical analysis been performed appropriately and rigorously? 

Reviewer #1: N/A

Reviewer #2: N/A

3. Have the authors made all data underlying the findings in their manuscript fully available?

Reviewer #1: Yes

Reviewer #2: Yes

4. Is the manuscript presented in an intelligible fashion and written in standard English?

Reviewer #1: Yes

Reviewer #2: Yes

5. Review Comments to the Author

**Reviewer #1:**  Dear authors,

I would like to commend you on the work, particularly for the detailed and methodological care taken in the study. Below, I have a few comments that I believe may further enhance the manuscript.

The focus of the study is appropriate, as it highlights that, in addition to intense challenges, caregivers may find positive meanings in their experiences, as indicated by previous studies. However, there is a point that could be refined. The phrase "positive aspects may mitigate negative outcomes" can be interpreted in a superficial and potentially risky manner. While the literature suggests that positive experiences may attenuate negative impacts, it would be advisable to add a caution about the importance of not romanticising caregiving, and to acknowledge that negative impacts are not necessarily eliminated by these positive aspects. Therefore, I suggest adjusting the tone of some statements that suggest positive aspects can mitigate the negative ones, recognising that both coexist without necessarily eliminating the negative impact of caregiving, particularly in crisis situations.

The methodology was well-developed, detailed, and coherent. I only have a few remarks: Regarding the statement, "The questions were validated by a professional caregiver to ensure clarity and relevance," how was this validation conducted? Was a cultural adaptation interview performed with this caregiver, or did they simply read the questions and provide feedback? How does this strategy validate the questionnaire? What is the theoretical justification for it?

The validation of qualitative interview guides is typically conducted through a "pilot study," where a cultural adaptation (the technically correct term for a “pilot study” in qualitative research) is carried out to assess the researcher’s approach and to test whether the guide effectively supports the data collection process for the study’s objectives. The authors should explain this.

What recruitment strategy was used for these professionals? What was the process for sending the consent forms and obtaining authorisations? Was it done virtually? Which platforms or software were used?

If possible, I suggest including a diagram or images to illustrate the coding categorisation process. This would reduce the textual explanation and enhance readers' understanding of the process.

On saturation: Saturation, in itself, is a construct that brings the formally described criteria in its concept. According to Glaser and Strauss, the authors of this concept applied to qualitative research, the criterion of saturation is automatically applied when researchers seek to “saturate” the understanding of the phenomenon being studied, to the point where the research objective is answered. Therefore, it is not a limitation of the study to not have a specific criterion; saturation is, in fact, the sample closure criterion.

The section on the Study Field could be added to the methodology. This could facilitate understanding of the results in the context within which the participants were situated, particularly as there are often important differences in healthcare work between public and private settings. This also presents an opportunity for the authors to justify: why include both public and private contexts? In terms of representativeness, it may not make sense to compare both in a qualitative study?

What is the rationale behind considering hairdressers and cleaners as caregivers? While the nature of both professions may indeed transform into caregiving, in its broader sense, the study’s justification refers to healthcare work as the subject. Including categories outside of the healthcare worker context might be questioned. How do the authors justify their inclusion in this context?

On the results: There is a fine line between optimism and denial in the participants' statements. On a "bad day," when many distressing things occur, it is natural to feel drained. Seeing something truly distressing as “small” does not seem to me like optimism, but rather denial.

I understand optimism as a rational process of perceiving the whole, beyond the micro-space of anguish. Recognising that the distressing moment naturally exists but should not be the sole factor influencing one’s energy to act. Denying the existence of anguish is not positivity. Some authors might refer to this as toxic positivity.

I suggest the authors better substantiate their analyses and interpretations based on more coherent and rigorous positivity frameworks, to avoid pitfalls such as this.

The theme of satisfaction, gratitude, and improvement in relationships is scattered across all categories. I do not see the need for a specific category solely addressing these elements, as they do not seem to have their own identity to justify being separate categories.

On the discussion: The discussion addresses positivity and resilience carefully, acknowledging the difficulties faced by caregivers without falling into the trap of romanticising suffering or presenting toxic positivity.

However, some sections, such as the description of "satisfaction derived from participants' roles," could benefit from a clearer explanation that these positive feelings do not eliminate the emotional burden of caregiving during the pandemic. This would ensure that positivity is not interpreted as a way to downplay suffering.

For this reason, I suggest delving more deeply into the distinction between healthy resilience and toxic positivity. While the cited literature extensively discusses resilience and post-traumatic growth, it would be valuable to acknowledge that, in some cases, these experiences may coexist with unresolved psychological distress, something that should be addressed to avoid superficial interpretations.

Overall, incorporating a deeper analysis of how caregivers balanced resilience with emotional burden would ensure that the discussion remains critical and grounded in the reality of lived experiences.

**Reviewer #2: ** Thank you for giving me the opportunity to review this paper: Positive outcomes among Nursing Home Caregivers in Spain during the COVID-19 pandemic: A qualitative interview study

This paper meets the criteria for publication in PLOS ONE. It reports important findings regarding positive outcomes among nursing home caregivers during the COVID- pandemic.

I have some comments and suggestions connected to the introduction and method that need some further development.

Introduction: Due to the data collected from different professional groups in the nursing homes, more supplementary information about the context of Spanish nursing homes, how they were organized, and work assignments need to be included. A more complementary context will also give a greater understanding of how their collaboration across positions contributed to positive outcomes, which is the article's purpose. Furthermore, we know the restrictions and measures implemented differed in different countries. A brief description of what this was like during COVID-19 will also contribute to an increased understanding of the results.

Method: An account is given of how informants were recruited and why having different occupational groups within the nursing homes has been desirable. It is pointed out that by interviewing different employees who have different work functions, we will get a more holistic understanding of what has been a positive outcome. It is argued that: This approach allowed for a richer and more inclusive analysis, reflecting the varied roles and experiences of the entire caregiving team.

Mixing different employee positions and then functions can at the same time, create ambiguities in the further reading of the analysis and results because their roles involve different types of responsibilities, tasks and physical proximity to sick residents, contact with relatives etc. (see the comment above about the introduction)

It is pointed out that themes mentioned by less than 40% were seen as non-saturation and, therefore, not included. This opens up the question of whether the positive outcomes were more or less equally distributed across the employees or whether some occupational groups stood out. If possible, indicating in the text if there were differences would have increased the quality of the analysis and the results.

Although the study aims to illuminate positive outcomes, the interview guide has also asked for challenges and negative impacts that emerge as data in the Coding manual (supplementary material 2). These categories, the results, could have advantageously been highlighted explicitly both in the introduction as part of the body of former research on the challenges during the COVID-19 pandemic and in the discussion section.

You have mixed reference styles in the paper. The list of references is numbered in Vancouver style, but in some places in the text, in the analysis and discussion section, you have included the references by name and date without a number.

I hope these suggestions will be helpful as you refine your paper. Thank you for the opportunity to review it.

6. PLOS authors have the option to publish the peer review history of their article (what does this mean? ). If published, this will include your full peer review and any attached files.

**Do you want your identity to be public for this peer review?** For information about this choice, including consent withdrawal, please see our Privacy Policy .

Reviewer #1: No

Reviewer #2: No

---

## [Author Response · Author response to Decision Letter 1]

5 Dec 2024

Response provided in the attached document

---

## [Decision Letter · Decision Letter 1]

24 Feb 2025

Positive outcomes among nursing home caregivers in Spain during the COVID-19 pandemic: A qualitative interview study

PONE-D-24-43127R1

Dear Dr.. Ordóñez-Carabaño ,

We’re pleased to inform you that your manuscript has been judged scientifically suitable for publication and will be formally accepted for publication once it meets all outstanding technical requirements.

Kind regards,

Rosemary Frey

Academic Editor

PLOS ONE

Additional Editor Comments (optional):

Reviewers' comments:

Reviewer's Responses to Questions

**Comments to the Author**

1. If the authors have adequately addressed your comments raised in a previous round of review and you feel that this manuscript is now acceptable for publication, you may indicate that here to bypass the “Comments to the Author” section, enter your conflict of interest statement in the “Confidential to Editor” section, and submit your "Accept" recommendation.

Reviewer #1: All comments have been addressed

Reviewer #2: All comments have been addressed

Reviewer #3: (No Response)

2. Is the manuscript technically sound, and do the data support the conclusions?

Reviewer #1: Yes

Reviewer #2: Yes

Reviewer #3: Yes

3. Has the statistical analysis been performed appropriately and rigorously? 

Reviewer #1: N/A

Reviewer #2: N/A

Reviewer #3: N/A

4. Have the authors made all data underlying the findings in their manuscript fully available?

Reviewer #1: Yes

Reviewer #2: Yes

Reviewer #3: Yes

5. Is the manuscript presented in an intelligible fashion and written in standard English?

Reviewer #1: Yes

Reviewer #2: Yes

Reviewer #3: Yes

6. Review Comments to the Author

Reviewer #1: I consider the authors' responses and justifications to be well-founded, and the revisions implemented have significantly enhanced the clarity and impact of the manuscript. The improvements have strengthened its scientific contribution, making it even more compelling and insightful. Congratulations to the authors on their diligent efforts.

Reviewer #2: Thank you for the revised article. You have responded and implemented the comments in a satisfactory manner.

Reviewer #3: The authors use an appropriate methodology and describe their findings regarding the experiences and learning that nursing home caregivers in Spain experienced in coping with the COVID-19 pandemic: post-traumatic growth, cohesive team with improvement in relationships, resilience, finding meaning, job satisfaction...

Apart from experiencing it in my natural workplace, I had the opportunity to collaborate with nursing home workers during the COVID-19 pandemic and my perception is that this work describes this experience very adequately.

7. PLOS authors have the option to publish the peer review history of their article (what does this mean? ). If published, this will include your full peer review and any attached files.

**Do you want your identity to be public for this peer review?** For information about this choice, including consent withdrawal, please see our Privacy Policy .

Reviewer #1: **Yes: ** Rodrigo Almeida Bastos

Reviewer #2: No

Reviewer #3: **Yes: ** Álvaro Sanz Rubiales

---

## [Editor Report · Acceptance letter]

PONE-D-24-43127R1

PLOS ONE

Dear Dr. Ordóñez-Carabaño,

I'm pleased to inform you that your manuscript has been deemed suitable for publication in PLOS ONE. Congratulations! Your manuscript is now being handed over to our production team.

Kind regards,

on behalf of

Dr. PLOS Manuscript Reassignment

Staff Editor

PLOS ONE
